# Millimeter-Wave Channel Modeling in a Vehicular Ad-Hoc Network Using Bose–Chaudhuri–Hocquenghem (BCH) Code

**Arshee Ahmed** [1,*,†] **, Haroon Rasheed** [1,†] **and Madhusanka Liyanage** [2,3]

1   Electrical Engineering Department, Bahria University, Karachi 75260, Pakistan; haroonrasheed.bukc@bahria.edu.pk
2   School of Computer Science, University College Dublin, D04 V1W8 Dublin, Ireland; madhusanka@ucd.ie
3   Centre for Wireless Communications, University of Oulu, 90014 Oulu, Finland
*   Correspondence: 02-281171-001@student.bahria.edu.pk
†   These authors contributed equally to this work.

**Abstract:** The increase in capacity demand for vehicular communication is generating interest among researchers. The standard spectra allocated to VANET tend to be saturated and are no longer enough for real-time applications. Millimeter-wave is a potential candidate for VANET applications. However, millimeter-wave is susceptible to pathloss and fading, which degrade system performance. Beamforming, multi-input multi-output (MIMO) and diversity techniques are being employed to minimize throughput, reliability and data rate issues. This paper presents a tractable channel model for VANET in which system performance degradation due to error is addressed by concatenated Alamouti space-time block coding (ASTBC) and Bose–Chaudhuri–Hocquenghem (BCH) coding. Two closed-form approximations of bit error rate (BER), one for BCH in Rayleigh fading and the second for BCH with ASTBC, are derived. These expressions comprise SNR and code rate and can be utilized in designing VANET architectures. The results show that the BER using concatenated ASTBC and BCH outmatches the conventional BER ASTBC expression. The analytical results are compared with numerical results, thereby showing the accuracy of our closed-form expressions. The performance of the proposed expressions is evaluated using different code rates.

**Keywords:** millimeter-wave; VANET; BCH coding; Alamouti space-time coding; bit error rate

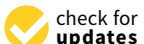

## 1. Introduction

People die and get injured in traffic accidents all over the world. It is mandatory for all vehicles to get information about traffic and road conditions to avoid accidents [1]. To cater to this issue, an effective and efficient transport management system is required to minimize road accidents and congestion, as specified in [2]. It is mentioned in [3] that 60 percent of accidents can be avoided if warning messages are delivered to drivers before accidents. Hence, scientists have introduced vehicular ad-hoc networks (VANETs), which turn every vehicle into a wireless node that can communicate with other vehicles. The VANET is a special class of mobile ad-hoc network (MANET) formed by vehicles equipped with wireless gadgets. VANET communication takes place between vehicle to vehicle (V2V) mode and vehicle to roadside unit (RSU), forming an intelligent transport system as shown in Figure 1. The Wireless Access Vehicular Environment (WAVE) was adopted from 802.11a and is used in intelligent transportation systems (ITS). In VANET, vehicles communicate with each other to share information required for safety, comfort, entertainment and low latency purposes.

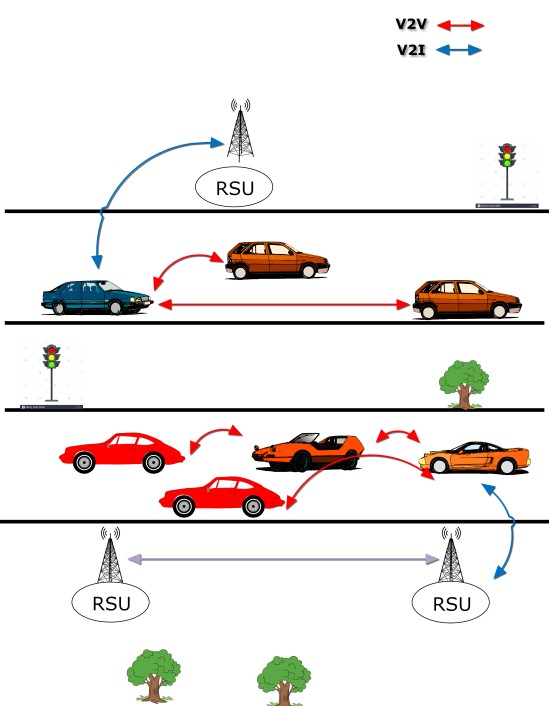

**Figure 1.** VANET architecture.

In Table 1, VANET communication technologies are summarized. Bluetooth, RFID and GPS are simplex types of communication. The other technologies are duplex. The range of Bluetooth is minimal, i.e., approximately 10 m. 2G/3G/4G/5G and GPS provide global coverage.

The demand for communication in VANET with high data rates has dramatically increased in recent years. This requires the utilization of limited resources such as power and bandwidth as efficiently as possible to support the massive communication demand in VANET. Currently, for wireless communication, millimeter-wave (mmWave) massive MIMO has been identified as the most promising technology for vehicular communication [4] The millimeter-wave band comprises a range from 30 to 300 GHz, which means a wavelength in between 1 and 10 mm. Millimeter-wave has lots of benefits and advantages, due to which it is suitable for 5G communication, cellular communication and other wireless communication. High accuracy, small antenna size, smaller cell and power efficiency are some advantages of millimeter wave.

**Table 1.** VANET communication technologies.

| Technology | Type | Distance |
|:---:|:---:|:---:|
| RFID | simplex | 10 m |
| Bluetooth | simplex | ≈10 m |
| WiFi | duplex | ≈50 m |
| WAVE/802.11p DSRC | duplex | ≈1 km |
| 2G/3G/4G/5G | duplex | global |
| GPS | simplex | global |
| WiMAX | duplex | ≈1 km |
| ZigBee | duplex | ≈20 m |

Due to the high mobility of vehicular nodes and the multipath fading, reliability, throughput and accuracy are the important aspects of VANET millimeter-wave communication. In millimeter-wave communication, the signal distorts and attenuates more due to the short wavelength. Pathloss is very high because of high career frequency. Further, it is

more susceptible to rain and atmospheric attenuation. In past research, MIMO, space-time block coding (STBC) and cooperative MIMO have been tested to achieve reliability and a high data rate. MIMO using ASTBC along with error control coding has not been explored yet. For vehicular communication, a manipulable architecture is needful which provides us with high reliability and a lower bit error rate.

The main contribution of our work is the derivation of exact closed-form expressions to calculate the BER using ASTBC and BCH coding. Our derived expressions can be utilized in BER computations, in contrast to SNR and code rate. Further, SNR can be computed for two blockers, which ultimately gives us an error probability for two blockers. The results show that our proposed BER outperforms the traditional ASTBC BER equation. Both numerical and simulation results verify each other, and hence provide a basic point of reference for further exploration. A significant reduction in computational complexity was obtained when evaluating system performance using our derived expressions.

The organization of the paper is as follows. Related work is explained in Section 2. Section 3 defines the ASTBC in which signal transmission using two transceivers is described. Section 4 describes the system model in which the pathloss of our model is computed. The closed-form approximations of BER for BCH in Rayleigh fading and BCH-concatenated ASTBC coding are derived. Section 5 contains results and discussion. Last comes the conclusion in Section 6.

## 2. Related Work

Currently, Dedicated Short-Range Communication (DSRC) is a VANET communication protocol. The coverage range of DSRC is about 1 km, with achievable data rates in the order of 2–6 Mbps. The massive data rate that is expected to be required by the next generation of automotive applications cannot be achieved using these technologies [5]. A multi-band orthogonal frequency division multiplexing (MB-OFDM)-based ultra-wideband technique is used to transmit large data blocks in VANET [6]. Various studies have been carried out regarding bit error analysis. BER was studied on mobile station MS velocity variations using 802.11p. Results showed that there is very little impact from velocity variations on bit error performance. However, when the data frame length or relative speed increases, performance is affected. Millimeter-wave is a good solution for high-speed Internet access. The power delay profile (PDP) of the wireless intra-vehicular channel was derived from 1000 data sets. A tapped-delay-line (TDL) was derived based on the proposed PDP profile. The TDL model was used for a bit error study of millimeter-wave by varying data rates and link length. Since millimeter waves cannot penetrate walls, a relay device is required to redirect signals [7]. In [8] bit error rate probability was evaluated for interference in a multi-hop millimeter-wave system. To achieve reliability and continuity in vehicular communication, an adaptive transmission mechanism was proposed. The mechanism should adjust the data rate according to the number of connected vehicles [9].

The current communication technologies cannot meet the high data rate requirements in vehicular communication. Therefore, a radio on fiber architecture was proposed [10]. To achieve a high data rate, a higher modulation order is used in radio on fiber architecture. An adaptive modulation mechanism was proposed [11]. An architecture of VANET using multi-gigabit communication was proposed in [12]. In the proposed architecture, i.e., the Giga-V2V (GiV2V) network, millimeter-wave is used to deliver high-quality video and sensor data of smart and self-driving cars. A few years back, only wireless access in the vehicular environment (WAVE)/IEEE 802.11p was available for V2V communication. New release 14 of long term evolution (LTE) and the arrival of 5G opened up new challenging possibilities. Due to the widespread deployment of the cellular systems and low latency, LTE V2V supplanted WAVE/IEEE 802.11p. However, LTE V2V is in the preliminary phase. There are certain drawbacks in current communication technologies. In WAVE/IEEE 802.11p, performance degrades with an increase in channel load. In high density areas, collisions occur, due to which network performance degrades in terms of throughput and latency [13].

## 3. Alamouti Code (Transmitter Diversity)

This section gives an overview of ASTBC. In ASTBC, two transmitters and two receivers are used. Consider a vehicular node i equipped with two transmitting antennas $tx_0$ and $tx_1$, and node j equipped with two receiving antennas $rx_0$ and $rx_1$. The channel between $tx_0$ and $rx_0$ is denoted by $h_0$; that between $tx_1$ and $rx_0$ is denoted by $h_1$. $h_2$ represents the channel between $tx_0$ and $rx_1$ and $h_3$ represents the channel between $tx_1$ and $rx_1$, as shown in Figure 2.

The two received signals in two adjacent symbol intervals can be written as

$$r_0 = h_0.s_0 + h_1.s_1 + n_0$$

$$r_1 = -h_0.s_1^* + h_1.s_o^* + n_1$$

$$r_2 = h_2.s_0 + h_3.s_1 + n_2$$

$$r_3 = -h_2.s_1^* + h_3.s_o^* + n_3$$

where $n_1$, $n_2$, $n_3$ and $n_4$ are complex thermal noise and interference. The receiver $rx_0$ receives two symbols $r_0$ and $r_1$ in time t and the receiver $rx_1$ receives two symbols $r_2$ and $r_3$ in time t + T.

The combiner combines signals and gives two O/P signals $y_0$ and $y_1$.

$$y_0 = h_0^*.r_0 + h_1.r_1^* + h_2^*.r_2 + h_3.r_3^*$$

$$y_1 = h_1^*.r_0 - h_0.r_1^* + h_3^*.r_2 - h_2.r_3^*$$

$$y_0 = (\alpha_0^2 + \alpha_1^2 + \alpha_2^2 + \alpha_3^2).s_0 + h_0^*.r_0 + h_1.r_1^* + h_2^*.r_2 + h_3.r_3^*$$

$$y_1 = (\alpha_0^2 + \alpha_1^2 + \alpha_2^2 + \alpha_3^2).s_1 + h_1^*.r_0 - h_0.r_1^* + h_3^*.r_2 - h_2.r_3^*$$

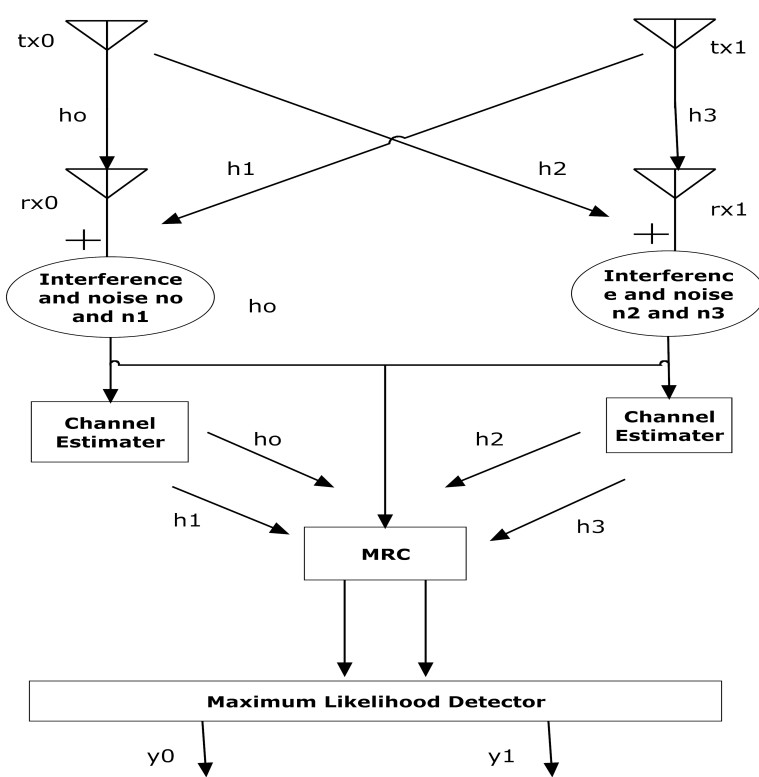

**Figure 2.** Alamouti space-time coding.

## 4. System Model

A vehicular network with N number of nodes is considered for modeling our propagation channel. d is the transmission range in meters. Figure 3 depicts the simulation model in which node i is communicating with node j. Consider a scenario in which node i is approaching node j which is decelerating. Node j will send a warning message to node i. We have simulated the scenario and have compared our results with simulated BER in the Rayleigh channel. Our BER computational approach with node i and node j is described in the next subsections. The simulations were performed in Matlab, as Matlab is user-friendly, and developing computational code and plotting graphs is easy in Matlab.

The pathloss (PL) in db can be computed using expression [14]

$$PL = 10.\alpha.log_{10}(d) + \frac{15.d}{1000} + \beta \tag{1}$$

where $\alpha$ represents path loss exponent; $\beta$ corresponds to a parameter for different numbers of blockers. The middle term in Equation (1) represents atmospheric attenuation for 60 GHz. The simulation parameters are shown in Table 2.

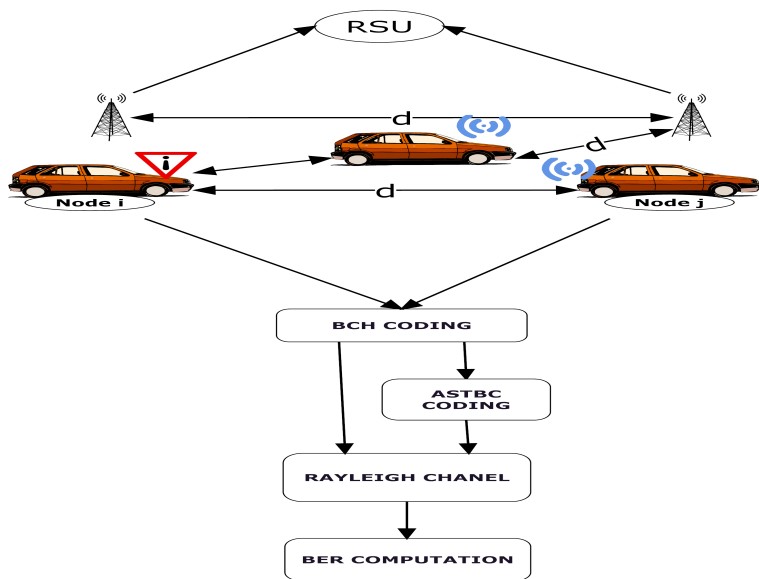

**Figure 3.** Simulation model.

**Table 2.** Pathloss simulation parameters.

| n | $\beta$ | $\alpha$ |
|---|---|---|
| 0 | 70 | 1.77 |
| 1 | 78.6 | 1.71 |
| 2 | 115 | 0.635 |

The received SNR $\gamma$ can be computed using expression (2).

$$\gamma = P_t + G_{tx} + G_{rx} - NP - PL - N_t \tag{2}$$

where $P_t$ corresponds to transmit power, $G_{tx}$ represents transmitter array gain, $G_{rx}$ represents receiver array gain, $N_t$ represents the number of transmitters, $R_c$ corresponds to code rate, $NF$ represents noise figure and NP corresponds to noise power which is computed using the expression below.

$$NP = -174 + NF$$

Simulation parameters of SNR are depicted in Table 3.

**Table 3.** SNR simulation parameters.

| Parameter | Value |
| --- | --- |
| $P_t$ | 10 dB |
| $G_{tx}$ | 20 dB |
| $G_{rx}$ | 20 dB |
| $N_t$ | 2 |
| $R_c$ | 0.9 |
| NF | 6 dB |

These simulation parameters were used to compute the SNR of our model. In Sections 4.1 and 4.2, the SNR is used in our closed-form expression derivation.

*4.1. Bose–Chaudhuri–Hocquenghem (BCH) Coding in Rayleigh Fading on Node i*

In this subsection, a closed-form expression for BCH coding in Rayleigh fading is derived. The probability of error of BCH coding in Rayleigh fading on node i can be written as

$$P_{ei} = \int_0^{+\infty} P_{BCH}.p_\gamma.d\gamma \tag{3}$$

where

$$P_{BCH} = Q.\sqrt{2.R_c.\gamma} \tag{4}$$

and,

$$p_\gamma = \frac{1}{\bar{\gamma}}.exp^{-\frac{\gamma}{\bar{\gamma}}} \tag{5}$$

Substituting expression (4) and expression (5) into expression (3) yields,

$$P_{ei} = \int_0^{+\infty} (Q\sqrt{2.Rc.\gamma}).\frac{1}{\bar{\gamma}}.exp^{-\frac{\gamma}{\bar{\gamma}}}d\gamma \tag{6}$$

In [15] (Equations (A.35) and (A.36), pp. 114, 134)

$$Q\sqrt{2.x} = Q(x)$$

Using this substitution,

$$Q\sqrt{2.(2.\gamma.Rc)} = Q(2.Rc\gamma)$$

where $x = Q(2.Rc.\gamma)$ Substituting the above expression in expression (6) yields

$$P_{ei} = \int_0^{+\infty} Q(2.Rc.\gamma).\frac{1}{\bar{\gamma}}.exp^{-\frac{\gamma}{\bar{\gamma}}}d\gamma \tag{7}$$

Converting $Q$ function into an error function gives [16] (Equation (2.55), p. 56)

$$Q(x) = \frac{1}{2}.erfc(\frac{x}{\sqrt{2}})$$

where $x = (2.Rc.\gamma)$ Substituting x in above expression gives us

$$Q(2.Rc.\gamma) = \frac{1}{2}.erfc((2.R_c.\gamma)/\sqrt{(2)})$$

Substituting the above expression into Equation (7) gives

$$P_{ei} = \int_0^{+\infty} \frac{1}{\bar{\gamma}}.exp^{-\frac{\gamma}{\bar{\gamma}}}.\frac{1}{2}.erfc(2.R_c.\gamma/\sqrt{2}) \tag{8}$$

Applying integration by parts gives

$$P_e = \frac{1}{\bar{\gamma}}.exp^{-\frac{\gamma}{\bar{\gamma}}}.\int_0^{+\infty} \frac{1}{2}.erfc(2.R_c.\gamma/\sqrt{2}) -$$
$$\int_0^{+\infty} \frac{d}{d\gamma}(\frac{1}{\bar{\gamma}}.exp^{-\frac{\gamma}{\bar{\gamma}}}).\int_0^{+\infty} \frac{1}{2}.erfc(2.R_c.\gamma/\sqrt{2}) \tag{9}$$

In [17] (Equation (3), p. 4)

$$\int_0^{+\infty} erfc(ax)dx = \frac{1}{a.\sqrt{pi}}$$

where $a = \frac{2.R_c}{\sqrt{2}}$ Therefore,

$$\int_0^{+\infty} \frac{1}{2}.erfc(2.R_c.\gamma/\sqrt{2}) = \frac{1}{2}.\frac{1}{\sqrt{pi}.R_c.2/\sqrt{2}} \tag{10}$$

Plugging expression (10) in expression (9) gives us

$$P_{ei} = (\frac{1}{\bar{\gamma}}.exp^{-\frac{\gamma}{\bar{\gamma}}}.\frac{1}{2}.\frac{1}{\sqrt{pi}.R_c.2/\sqrt{2}}) -$$
$$\int_0^{+\infty} (\frac{d}{d\gamma}.\frac{1}{\bar{\gamma}}.exp^{-\frac{\gamma}{\bar{\gamma}}}).\frac{1}{2}.\frac{1}{\sqrt{pi}.R_c.2/\sqrt{2}} \tag{11}$$

Applying differentiation gives

$$P_{ei} = (\frac{1}{\bar{\gamma}}.exp^{-\frac{\gamma}{\bar{\gamma}}}.\frac{1}{2}.\frac{1}{\sqrt{pi}.R_c.2/\sqrt{2}}) -$$
$$\int_0^{+\infty} (-\frac{1}{\bar{\gamma}^2}.exp^{-\frac{\gamma}{\bar{\gamma}}}).\frac{1}{2}.\frac{1}{\sqrt{pi}.R_c.2/\sqrt{2}} \tag{12}$$

In [18] (Equation (3.310), p. 334):

$$\int_0^{+\infty} exp^{-\frac{\gamma}{\bar{\gamma}}} = 1/\frac{1}{\bar{\gamma}}$$

Plugging the above expression in Equation (12) yields

$$P_{ei} = (\frac{1}{\bar{\gamma}}.exp^{-\frac{\gamma}{\bar{\gamma}}}.\frac{1}{2}.\frac{1}{\sqrt{pi}.R_c.2/\sqrt{2}}) + \frac{1}{\bar{\gamma}^2}.\frac{1}{1/\bar{\gamma}}.\frac{1}{2}.\frac{1}{\sqrt{pi}.R_c.2/\sqrt{2}} \tag{13}$$

Hence, the closed form expression of BCH probability of error in terms of mmWave Rayleigh fading can be written as

$$P_{ei} = \frac{1}{\bar{\gamma}}.exp^{-\frac{\gamma}{\bar{\gamma}}}.\frac{1}{2}.\frac{1}{\sqrt{pi}.R_c.2/\sqrt{2}} + \frac{1}{\bar{\gamma}}.\frac{1}{2}.\frac{1}{\sqrt{pi}.R_c.2/\sqrt{2}} \tag{14}$$

The bit error rate was computed for different values of the code rate, i.e., (127,64), (127,36) and (255,251) using Equation (14).

### 4.2. BER in Concatenated ASTBC and BCH on Node j

In this section the closed-form expression of BER of ASTBC using BCH coding is derived. The probability of error on Node j using ASTBC and BCH can be written as

$$P_{ej} = \int_0^{+\infty} P_{BCH}.P_{ASTBC}.d\gamma \tag{15}$$

where $P_{BCH}$ is given in expression (4). The probability of error in ASTBC ($P_{ASTBC}$) can be written as

$$P_{\text{ASTBC}} = \frac{1}{2} - \frac{1}{2}.(1 + \frac{1}{\gamma})^{-1/2}$$

Plugging above expression and expression (4) in expression (15) yields

$$P_{ej} = \int_0^{+\infty} Q\sqrt{2.R_c.\gamma}.(\frac{1}{2} - \frac{1}{2}.(1 + \frac{1}{\gamma})^{-1/2})d\gamma \tag{16}$$

Using the substitution used in previous section where

$$Q\sqrt{2.x} = Q(x)$$

in expression (16) can be written as

$$P_{ej} = \int_0^{+\infty} Q(2.R_c.\gamma).(\frac{1}{2} - \frac{1}{2}.(1 + \frac{1}{\gamma})^{-1/2})d\gamma \tag{17}$$

Converting Q function into the error function yields

$$P_{ej} = \int_0^{+\infty} \frac{1}{2}.erfc(\frac{2.R_c.\gamma}{\sqrt{2}}).(\frac{1}{2} - \frac{1}{2}.(1 + \frac{1}{\gamma})^{-1/2})d\gamma$$
$$= \int_0^{+\infty} \frac{1}{4}.erfc(\frac{2.R_c.\gamma}{\sqrt{2}}) - \int_0^{+\infty} (\frac{1}{4}erfc(\frac{2.R_c.\gamma}{\sqrt{2}}). \tag{18}$$
$$(1 + \frac{1}{\gamma})^{-1/2})d\gamma$$

By using the relation described in expression (10), i.e.,

$$\int_0^{+\infty} \frac{1}{2}.erfc(2.R_c.\gamma/\sqrt{2}) = \frac{1}{2}.\frac{1}{\sqrt{pi}.R_c.2/\sqrt{2}}$$

expression (18) can be written as

$$Pej = \frac{1}{4}.\frac{1}{\sqrt{pi}.R_c.2/\sqrt{2}} - \frac{1}{4}.\frac{1}{\sqrt{pi}.R_c.2/\sqrt{2}}.(1 + \frac{1}{\gamma})^{-0.5} \tag{19}$$

Expression (19) gives us a new tractable closed-form expression of BCH in ASTBC.

## 5. Results and Discussion

In this section results of the pathloss model and BER closed-form expressions are described. Figure 4 displays our pathloss for two vehicles (obstacles). It can be remarked with more obstacles, i.e., n = 0, 1 and 2, pathloss also increased. Further, when increasing distance between tx and rx, pathloss also rose. When the number of obstacles was 2, pathloss was −135 dB at 30 m.

Figure 5 shows the results of BCH in the Rayleigh fading using equation 14. The results are compared with the AWGN channel and the simulated BER in the Rayleigh channel. From Figure 5 it can be noticed that the BER was getting low when EbNo was increasing. The former's code rate (127,36) was superior. In Figure 6 the results using the conventional ASTBC equation and our proposed expression, i.e., expression 19 (BER using ASTBC with BCH coding), are shown. The BER of ASTBC with BCH coding was minimum in contrast with the BER without BCH coding. The simulated BER in the Rayleigh channel and the BER without BCH coding were similar. The former's code rate (127,36) was superior. Table 4 shows that our BER computation approaches outmatched other techniques.

**Table 4.** ASTBC and VANET BER computation approaches.

| Sr No. | Paper | Approach | Comparison with Proposed Approximations |
|--------|-------|----------|------------------------------------------|
| 1 | [19] | MIMO-OFDM System using ASTBC | Low BER is obtained using proposed approximations |
| 2 | [20] | ASTBC Impact on VANET | The Performance is evaluated using different modulaton schemes than code rate. |
| 3 | [21] | BCH-STBC | Low BER is obtained using proposed approximations |
| 4 | [22] | Fast Frequency Hopping Orthogonal Frequency Division Multiplexing (FFHOFDM) Pre-Coding | The results of proposed approximations are optimized. |

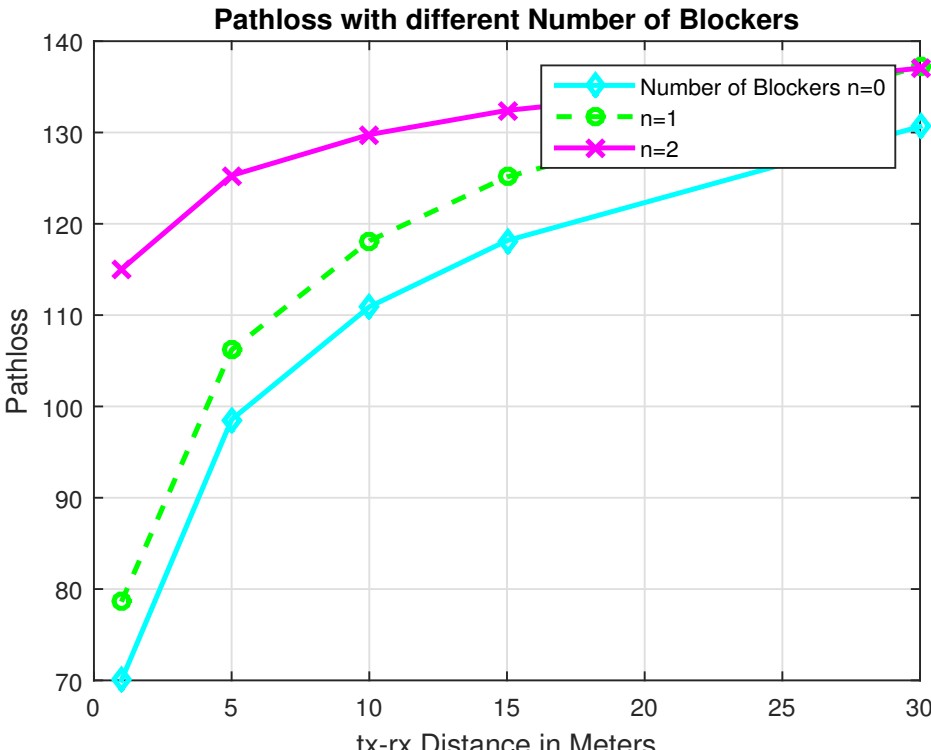

**Figure 4.** Pathloss in the presence of vehicles.

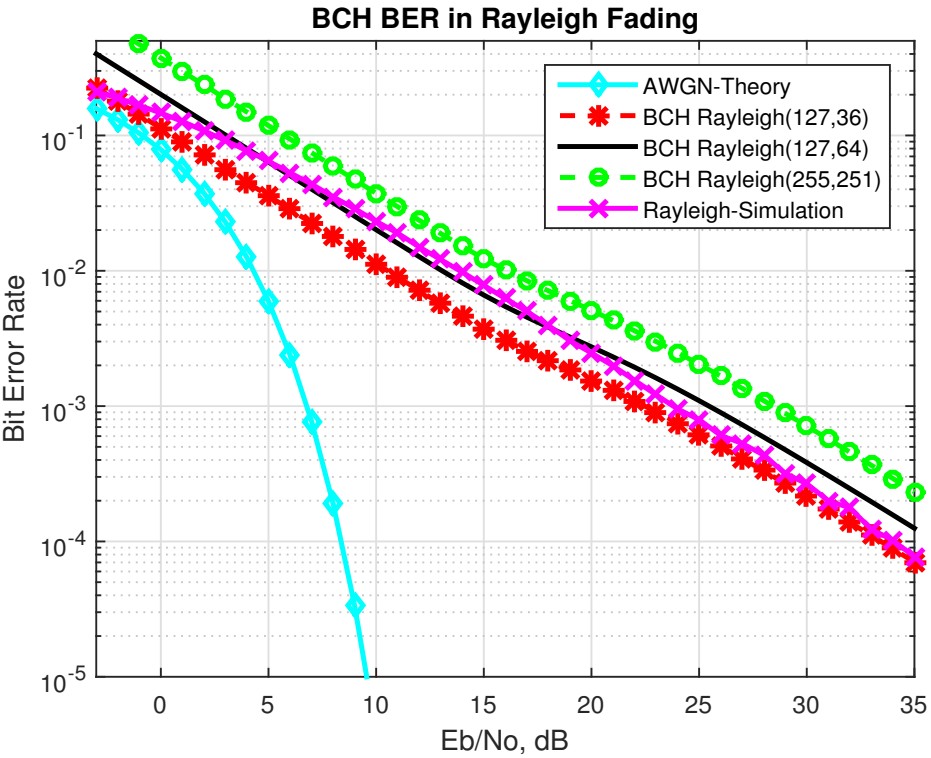

**Figure 5.** BER of BCH in Rayleigh fading.

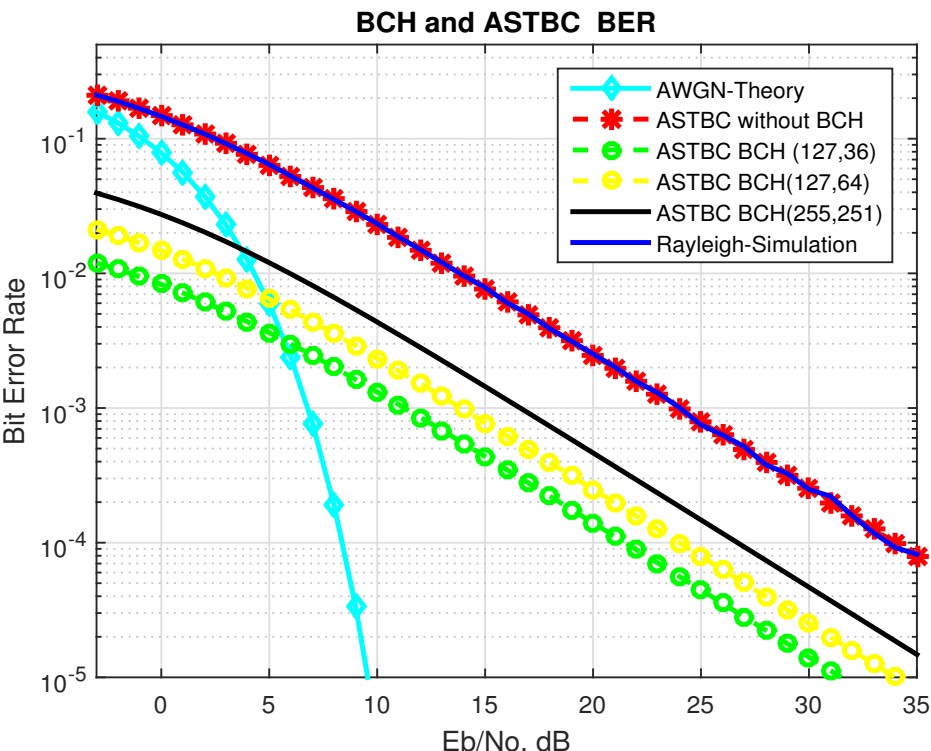

**Figure 6.** BER of BCH-ASTBC.

## 6. Conclusions

In this manuscript, a tractable channel model for millimeter-wave communication in VANET was presented. The main contribution of this research was to provide a statistical and analytical framework to evaluate the performance of VANET using millimeter-wave with channel coding, i.e., BCH and RS. The pathloss was computed for three vehicles. The pathloss was getting high in two cases—first the number of vehicles increased; secondly, the distance between tx and rx increased.

Using our derived closed-form expressions of BER, we simulated the BER of our proposed BCH in Rayleigh fading and the BER of concatenated ASTBC and BCH coding. The results showed that the BER using concatenated ASTBC and BCH coding was the most depleted. The result of the proposed approximation, i.e., expression 19, was compared with the orthodox BER ASTBC expression. The results showed that the BER using BCH-coded ASTBC was less than the BER using the orthodox expression for BER. The results were compared using different code rates. The BER using code rate 12,732 was the smallest.

The derived expressions were validated and verified for scenarios of practical interest and complemented by the simulation model. Furthermore, a significant reduction in the computational burden was obtained.

**Author Contributions:** Conceptualization, A.A. and H.R.; methodology, A.A.; validation, H.R. and M.L.; investigation, H.R.; writing—original draft preparation, A.A.; writing—review and editing, H.R. and M.L.; supervision, H.R. All authors have read and agreed to the published version of the manuscript.

**Funding:** This research received no external funding.

**Conflicts of Interest:** The authors declare that they have no conflict of interest.

## Abbreviations

The following abbreviations are used in this manuscript:

| | |
|---|---|
| VANET | Vehicular Ad-Hoc Network |
| OBU | On-Board Unit |
| RSU | Roadside Unit |
| BCH | Bose–Chaudhuri–Hocquenghem |
| ASTBC | Alamouti Space-Time Block Coding |
| BER | Bit Error Rate |
| ITS | Intelligent Transportation Systems |
| STBC | Space-Time Block Coding |
| DSRC | Dedicated Short-Range Communication |
| MB-OFDM | Multi-Band Orthogonal Frequency Division Multiplexing |
| PDP | Power Delay Profile |
| TDL | Tapped-Delay-Line |
| LTE | Long Term Evolution |

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
