# Peer review of "Millimeter-Wave Channel Modeling in a Vehicular Ad-Hoc Network Using Bose–Chaudhuri–Hocquenghem (BCH) Code"

_electronics, doi:10.3390/electronics10090992_

Round 1
Reviewer 1 Report
The presentation of the manuscript must be improved.
- The introductory section should not be divided into subsections.
- The title of section 2 should be Related Work.
- Section 2.1 should be an independent section.
- Description of the Method, Results and Discussions must be improved.
- There are several English mistakes. The manuscript must be checked by an English expert.
Reviewer 2 Report
The authors need to make some serious changes in order their work to be accepted.
- They need to provide information regarding the simulation platform that has been used as well as evidence about why they choose the specific one instead of others.
- Authors need to use a scenario that is close to reality and run it in order to get results.
- More KPI’s (Key Performance Indexes) should be addressed in order their approach more valuable.
- The authors approach should be compared and evaluated against other similar approaches and all performances should be plotted in bigger graphs than those that presented on this manuscript.
- Below the Figure 2 there are 7 equations in which some factors have an asterisk, which magnitude exactly depicted?
- In the plots should be used different styles such as dots and dashes or triangles instead of colors.
- In the conclusion section authors is good to provide what is their contribution to real world.
Round 2
Reviewer 2 Report
The authors considered all the suggestions that the reviewers made, and it's layout now is ready for publication.